# Heterogeneity in Utilization of Optical Imaging Guided Surgery for Identifying or Preserving the Parathyroid Glands—A Meta-Narrative Review

**DOI:** 10.3390/life12030388

**Published:** 2022-03-08

**Authors:** Eline A. Feitsma, Hugo M. Schouw, Milou E. Noltes, Wido Heeman, Wendy Kelder, Gooitzen M. van Dam, Schelto Kruijff

**Affiliations:** 1Department of Surgery, University Medical Center Groningen, University of Groningen, Hanzeplein 1, 9713 GZ Groningen, The Netherlands; e.a.feitsma@umcg.nl (E.A.F.); h.m.schouw@umcg.nl (H.M.S.); m.e.noltes@umcg.nl (M.E.N.); w.t.heeman@umcg.nl (W.H.); 2TRACER EUROPE BV, L.J. Zielstraweg 1, 9713 GX Groningen, The Netherlands; go@tracercro.com; 3Department of Surgery, Martini Hospital Groningen, Van Swietenplein 1, 9728 NT Groningen, The Netherlands; w.kelder@mzh.nl; 4Faculty Campus Fryslân, University of Groningen, Wirdumerdijk 34, 8911 CE Leeuwarden, The Netherlands; 5LIMIS Development BV, Henri Dunantweg 2, 8934 AD Leeuwarden, The Netherlands; 6Department of Nuclear Medicine and Molecular Imaging, University Medical Center Groningen, University of Groningen, Hanzeplein 1, 9713 GZ Groningen, The Netherlands

**Keywords:** optical imaging, fluorescence, thyroidectomy, parathyroid glands, hypoparathyroidism, near-infrared imaging, autofluorescence, indocyanine green angiography, laser speckle contrast imaging

## Abstract

Background: Postoperative hypoparathyroidism is the most common complication after total thyroidectomy. Over the past years, optical imaging techniques, such as parathyroid autofluorescence, indocyanine green (ICG) angiography, and laser speckle contrast imaging (LSCI) have been employed to save parathyroid glands during thyroid surgery. This study provides an overview of the utilized methods of the optical imaging techniques during total thyroidectomy for parathyroid gland identification and preservation. Methods: PUBMED, EMBASE and Web of Science were searched for studies written in the English language utilizing parathyroid autofluorescence, ICG-angiography, or LSCI during total thyroidectomy to support parathyroid gland identification or preservation. Case reports, reviews, meta-analyses, animal studies, and post-mortem studies were excluded after the title and abstract screening. The data of the studies were analyzed qualitatively, with a focus on the methodologies employed. Results: In total, 59 articles were included with a total of 6190 patients. Overall, 38 studies reported using parathyroid autofluorescence, 24 using ICG-angiography, and 2 using LSCI. The heterogeneity between the utilized methodology in the studies was large, and in particular, regarding study protocols, imaging techniques, and the standardization of the imaging protocol. Conclusion: The diverse application of optical imaging techniques and a lack of standardization and quantification leads to heterogeneous conclusions regarding their clinical value. Worldwide consensus on imaging protocols is needed to establish the clinical utility of these techniques for parathyroid gland identification and preservation.

## 1. Introduction 

Hypoparathyroidism is the most common complication of thyroid surgery. Transient hypoparathyroidism (lasting less than six months after surgery) has been reported up to 38%, and permanent hypoparathyroidism (lasting more than six months after surgery) up to 29% [1,2]. Patients experiencing hypoparathyroidism can develop fatigue, muscle cramps, bone pain, as well as severe seizures and cardiac arrhythmias [3]. In the case of permanent hypoparathyroidism, patients are at risk for kidney complications, increased mortality rates, and an impaired quality of life [4,5,6].

To prevent postoperative hypoparathyroidism, an increasing amount of research has been published in the past decade on perioperative optical imaging of the parathyroid glands during thyroid surgery. Optical imaging consists of imaging modalities that use the visible to near-infrared (NIR) spectrum of light facilitating intraoperative image acquisition without the use of harmful radiation [7]. Several optical imaging techniques aim to recognize and identify the parathyroid glands on the one hand, and on the other hand to estimate whether the parathyroid glands have been spared during thyroidectomy. If the parathyroid perfusion is assumed to be damaged, it is possible to autotransplant the gland. Parathyroid autotransplantation consists of implanting minced parathyroid gland material in the (sternocleidomastoid) muscle regaining vascularization, resulting in more than half of its function (48–96%) after 4–8 weeks, potentially preventing long-term postoperative hypoparathyroidism [8].

The three most commonly used optical imaging techniques are discussed in this article. The first technique aims to identify parathyroid glands by using autofluorescent properties. The physiology of these autofluorescent properties is still unknown, although the hypothesis is that the calcium-sensing receptor, which is more abundant in the parathyroid gland as compared to the surrounding tissue, could be the responsible fluorophore for the autofluorescence [9]. It should be noted that autofluorescence patterns in diseased parathyroid glands are different from those in healthy parathyroid glands [10]. Other structures, such as fatty tissue and electrocoagulated tissue, can also have different autofluorescence patterns, which can lead to false-positive results [11,12]. The second technique uses indocyanine green (ICG) fluorescence angiography to evaluate the glands’ vascularity. ICG is a fluorescent dye that specifically binds to plasma protein, making it possible to visualize perfusion with a NIR camera [13]. ICG is administered intraoperatively so that all highly vascularized tissues, including the parathyroid glands, become fluorescent. The third technique, laser speckle contrast imaging (LSCI), uses laser light to form a random interference pattern, the so-called speckle pattern. The movement of particles (i.e., red blood cells) in the tissue of interest causes a change in the speckle pattern correlating to blood flow. This change of pattern allows LSCI to visualize and quantify blood flow in real-time without the need for a fluorescent dye [14]. 

Previous reviews have been published on optical imaging of parathyroid glands in thyroid surgery, however, these reported various results regarding postoperative outcomes [15,16]. The purpose of this review is to discover what may explain these differences. In this study, we will focus only on healthy parathyroid glands during thyroid surgery in order to limit the heterogeneity of included studies. We hypothesize that the heterogeneous results might be explained by large differences in the utilized methods in current literature on this subject. Large differences in the application of the techniques and camera systems would most likely lead to such divergent results that they cannot be compared with each other. Therefore, this article outlines an overview of the aforementioned optical imaging techniques for identification and preservation of the parathyroid glands, focusing in detail on the methodology that was used in the literature.

## 2. Methods

### 2.1. Study Design

A meta-narrative approach was chosen for this article to provide an overview over the time of publications regarding the subject of optical imaging of parathyroid glands during total thyroidectomy. This review was performed according to the RAMESES standard (Realist And Meta-narrative Evidence Syntheses: Evolving Standards) [17]. 

### 2.2. Search Strategy

A search strategy was established to find publications using parathyroid autofluorescence, LSCI, or ICG-angiography during thyroidectomy to identify or preserve parathyroid glands. Searches were performed for “thyroidectomy” or related terms, as well as “autofluorescence”, “indocyanine green” or “laser speckle contrast imaging” and related words. PUBMED, EMBASE and Web of Science were searched (Appendix A: Search strategy).

### 2.3. Inclusion Criteria

Only English-written publications were included in this study. Studies investigating the detection or perfusion of the parathyroid glands during thyroidectomy were included. These studies should utilize one or more of the following techniques: autofluorescence for parathyroid gland identification, indocyanine green fluorescence angiography, or laser speckle contrast imaging to evaluate the perfusion status of the parathyroid glands. In this review, we chose to focus on original clinical studies, therefore, case reports with less than three patients undergoing thyroidectomy, reviews, meta-analyses, animal studies, and post-mortem studies were excluded after title and abstract screening. 

### 2.4. Processing of Data

To identify, analyze, and assess the appropriate publications, the artificial intelligence program ASReview v0.16 was used [18]. This software uses machine learning algorithms that are actively learning based on interactions with the researcher, and therefore increase screening efficiency. Inter-rater reliability was established at 97% by two independent researchers. Furthermore, the guideline produced by ASReview was followed to screen an additional 25% of the total number of search results after the last article that was marked as relevant by the software, to minimize the risk of missing additional relevant articles.

Duplicates were removed. The relevant articles were read in full text and processed using Mendeley reference manager [19]. The full process of selecting relevant publications is shown in Figure 1. Data that was extracted from the articles included authors, title, year, origin, the month of publication, study design, sample size, type of surgery, imaging technique and the ICG dose in the case of ICG-angiography. 

Each article was examined for the completion of five independent stages essential for professionalization and the clinical implementation of the aforementioned techniques. These stages were ranked from least strong to strongest, with a study not necessarily having to have completed all previous steps. An article can, for example, have met the first and third stages, while the second one is not fulfilled. It was subsequently indicated per an article in which stages were met. The stages were arranged as followed: Feasibility. In this stage, the aim of the study is to examine whether parathyroid glands can be identified or preserved by the technique used.Validation of the parathyroid gland’s identification. In this stage, recognition of the parathyroid glands with optical imaging is either validated by the surgeon, based on naked-eye recognition, or by histopathology. Histopathology was considered to be superior to optical recognition.Clinical consequences. Clinical choices are made in this stage based on the optical imaging technique that was used.Quantification of the fluorescence signal. The images are quantified in real-time or postoperatively at this stage.Standardization of the protocol. In this stage, the protocol is clearly standardized so that it could be performed at any location with roughly the same results. For each article, it was noted whether the following parts were named or standardized: camera-to-wound bed distance, camera angle, whether the overhead lights were turned off, and camera settings. Additionally, for ICG-angiography, the following was noted: the injection protocol, whether a video or an image was recorded, and the dose of injected ICG.

## 3. Results

The search resulted in 763 articles related to optical imaging in parathyroid glands. These articles were then screened using ASReview, after which 59 articles remained (Figure 1). Of these, 33 articles utilized autofluorescence in parathyroid imaging, 19 used ICG-angiography, 5 used a combination of both techniques, and 2 studied LSCI for parathyroid preservation. The large exclusion percentage is explained by the large number of animal studies, case reports and reviews related to the search terms. Figure 1 depicts the selection process.

In the following paragraphs, the applications of the techniques in the studies that we have analyzed are discussed one by one. Figure 2 provides an overview of the stages that the studies have reached. An overview of the variety of study designs within the articles can be found in Figure 3, which will be discussed in each of the paragraphs separately.

### 3.1. Autofluorescence

#### 3.1.1. Selected Articles

A total of 38 papers described the imaging of parathyroid glands by autofluorescence, published between June 2011 and December 2021. Table 1 describes a detailed description of the various studies.

#### 3.1.2. Study Population

The study population ranged from 5 to 542 patients, with a median of 56.5 patients per study. Most studies were performed in both open parathyroid surgery and open thyroid surgery (*n* = 19) [20,21,22,23,24,25,26,27,28,29,30,31,32,33,34,35,36,37,38], 17 studies analyzed open (total or partial) thyroid dissections alone [9,12,39,40,41,42,43,44,45,46,47,48,49,50,51,52,53], and 2 studies described endoscopic procedures on parathyroid glands or thyroid [54,55] (Table 1).

#### 3.1.3. Study Design

During the study analysis, large differences were seen in the design of the studies (Figure 2). Twenty-nine (76%) of the conducted studies were designed as prospective cohort studies, and five were retrospective data studies, including one international multicenter study. Furthermore, three randomized controlled trials (RCT) and one non-randomized controlled study (NRS) were performed (Table 1).

The parathyroid glands were only imaged in vivo in twenty-eight studies (74%), with the timing of imaging differing significantly between studies (Figure 3). For example, Kahramangil et al. (2017) described a study in which the imaging of parathyroid glands was performed immediately after the thyroid gland release and before the surgeon started actively looking for them. Their group then compared whether parathyroid glands could better be identified by autofluorescence than with the naked eye [51]. As in several other studies, a large RCT by Benmiloud et al. (2020) imaged the parathyroid glands immediately after the surgeon had searched for these glands with the naked eye. This study compared the intervention (NIR imaging) group with a control group in which they performed the standard procedure (the naked eye of the surgeon), to investigate whether NIR could contribute to the prevention of postoperative hypocalcemia [53]. 

Other studies visualized the parathyroid glands only ex vivo or utilized a combination of in and ex vivo. These studies focused on recognizing inadvertently removed parathyroid glands in the resected thyroid specimen in order to decide whether they should be auto transplanted [12,25,34,38,44,49]. An example of such an ex vivo imaging study is reported by Bellier et al. (2021), in which the thyroid specimen was examined ex vivo with a NIR camera for the presence of parathyroid glands, before sending the specimen to the pathologist for validation [44]. 

Only a few studies used pathological validation, mostly those that included parathyroidectomies [20,21,23,24,28,29,37]. However, in the study conducted by Liu et al. (2020), each suspicious parathyroid gland was biopsied and submitted for a frozen section analysis, after which the remaining part of the parathyroid gland was left in situ. Although other studies did not biopsy the parathyroid glands, as this could result in damaging them, no conclusions were drawn on postoperative PTH or calcium in this study [48]. The majority of studies performed an alternative validation by scoring the surgeon’s confidence in recognizing a parathyroid gland, for example by assigning their confidence with a score of 0, 1 or 2. This score was compared to the 0, 1 or 2 score assigned to the autofluorescence or to the quantified signal from the autofluorescence.

#### 3.1.4. Imaging Technique

In 2011, the first study was published, in which Paras et al. described that optical imaging of autofluorescence could be a simple surgical technique to identify parathyroid glands in the surrounding tissue in real-time. By using a spectroscopic technique in which only the tissue directly touching the fiber was analyzed for its autofluorescence characteristics, they found that parathyroid gland tissue had a two to eleven times higher fluorescence intensity compared to surrounding tissue at 820–830 nm [20]. Four other studies in our analysis have used such spectroscopy technology [9,21,22,29]. Over the years, many different systems have been developed to better assess the parathyroid glands. NIR cameras have the advantage over spectroscopy of catching the parathyroid, thyroid, and surrounding tissue autofluorescence in one view, being a full-field technology. However, the disadvantage is that it is currently not yet possible to quantify such datasets in real-time. Various applications of these cameras were developed, leading to some cases in which a fluorescent overlay could be placed over the white-light image. The most commonly used cameras were those from the manufacturers Karl Storz, Fluobeam, and PDE-NEO [20]. The spectroscope has developed towards the PTEye, a system with a fiber-optic probe that compares the thyroid and the parathyroid gland autofluorescence in real-time and provides feedback in both visual and auditory signals [29]. Three studies in this review have used this technique [29,30,37].

#### 3.1.5. Standardization of Protocol

Notable differences were observed in the standardization of the imaging protocol. Nearly all studies mentioned turning off the overhead operating theater lights when imaging was started to minimize the effect of ambient lighting (Figure 2). However, this was not done when using the PTEye device, as its results were not affected by ambient light due to its being a contact method. Most studies (*n* = 20 (53%)) did not describe the distance of the camera to the wound bed. In studies that used the spectroscope (*n* = 6), it was placed directly on the tissue, as described in most (*n* = 3 (43%)) of them. However, only 12 of the 31 studies (39%) using a NIR camera mentioned the camera-to-wound bed distance, ranging from a varying distance of 5 to 20 cm, with the camera not being placed on a tripod. The angle of the camera to the surgical incision plane under which the images were taken was not described. Only five studies (13%) reported camera settings, usually only being exposure time. In 11 of the included studies (29%), clinical decision making was made based on the findings obtained with the spectroscope or the camera. The fluorescence signal reported was not quantified in 23 studies (61%); in the other 15 studies, this was executed perioperatively (*n* = 7), postoperatively (*n* = 6), or both (*n* = 1).

#### 3.1.6. Study Outcomes

In general, most studies concluded that NIR imaging utilizing autofluorescence was an easy and safe real-time method to detect parathyroid glands in or ex vivo. Parathyroid glands were found more quickly (52% of glands were found before thorough inspection) and more often (98% vs. 90%) with NIR imaging than with the naked eye, both with the spectroscope and with the NIR camera [21,28,40]. Using NIR imaging, parathyroid glands that were not visible with the surgeon’s naked eye were also found in 11% of the dissected thyroid specimens [38,44]. Three large (n-)RCTs report a significant reduction in the rate of postoperative hypocalcemia in the NIR-group (5.2% vs. 20.9% *p* < 0.001 and 9.1% vs. 21.7%, *p* = 0.007) [39,40,53]. However, other large studies (one RCT, one prospective cohort and one retrospective cohort [41,42,47]) were unable to demonstrate a correlation in this regard (9.3% vs. 10.5%, *p* = 0.53) [47].

**Table 1 life-12-00388-t001:** Included articles utilizing autofluorescence. AF = autofluorescence, ICG = indocyanine green, PTx = parathyroidectomy, Tx = thyroidectomy, NIR = near infra red.

Application	First Author	Origin	Publicated (MM.YY)	Journal	Study Design	Surgery	Sample Size	Imaging Technique	In/Ex Vivo Imaging
AF	Paras [20]	USA	06.11	Journal of Biomedical Optics	Prospective cohort	PTx/Tx	21	Spectroscopy	In vivo
AF	McWade [9]	USA	12.13	Surgery	Prospective cohort	Tx	45	Spectroscopy	In vivo
AF	McWade [22]	USA	12.14	Journal of Clinical Endocrinology and Metabolism	Prospective cohort	PTx/Tx	116	Spectroscopy and NIR camera (Karl Storz)	In vivo
AF	McWade [21]	USA	01.16	Surgery	Prospective cohort	PTx/Tx	137	Spectroscopy	In vivo
AF	Falco [23]	Argentina	08.16	Journal of the American College of Surgeons	Prospective cohort	PTx/Tx	28	NIR camera (Fluobeam)	In vivo
AF	De Leeuw [24]	France	09.16	World Journal of Surgery	Prospective cohort	PTx/Tx	63	NIR camera (Fluobeam)	In and ex vivo
AF	Kim [50]	South Korea	12.16	Journal of Clinical Endocrinology and Metabolism	Prospective cohort	Tx	8	Digital camera and NIR diode + illuminator	In vivo
AF	Shinden [25]	Japan	06.17	World Journal of Surgery	Prospective cohort	PTx/Tx	17	NIR camera (PDE-Neo)	In and ex vivo
AF	Ladurner [54]	Germany	08.17	Surgical Endoscopy	Prospective cohort	Laparoscopic PTx/Tx	30	Endoscopy NIR camera (Karl Storz)	in vivo
AF	Falco [26]	Argentina	09.17	Surgical Endoscopy	Retrospective review of prospective data	PTx/Tx	74	NIR camera, unknown system	In vivo
ICG + AF	Kahramangil [51]	USA	12.17	Gland Surgery	Prospective cohort	Tx	44	NIR camera (Pinpoint for ICG, Fluobeam for AF)	In vivo
AF	Ladurner [55]	Germany	01.18	Annals of the Royal College of Surgeons of England	Prospective cohort	Laparoscopic Tx	20	Endoscopy NIR camera (Karl Storz)	In vivo
AF	Benmiloud [39]	France	01.18	Surgery	Non-randomized controlled study	Tx	513	NIR camera (Fluobeam)	In vivo
AF	Kim [52]	South Korea	02.18	Journal of the American College of Surgeons	Prospective cohort	Tx	38	Digital camera and NIR diode + illuminator	In vivo
ICG + AF	Alesina [27]	Germany	03.18	Langenbeck’s Archives of Surgergy	Prospective cohort	PTx/Tx	5	Endoscopy NIR camera (Karl Storz)	In vivo
AF	Kahramangil [28]	USA	04.18	Annals of Surgical Oncology	International multicenter retrospective cohort	PTx/Tx	210	NIR camera (Fluobeam)	In vivo
AF	Thomas [29]	USA	11.18	Thyroid	Prospective cohort	PTx/Tx	197	Spectrometer and Pteye	In vivo
AF	Thomas [30]	USA	01.19	Surgery	Prospective cohort	PTx/Tx	41	Pteye and Overlay Tissue Imaging System	In vivo
AF	Dip [40]	Argentina	05.19	Journal of the American College of Surgeons	RCT	Tx	170	NIR camera (Fluobeam)	In vivo
ICG + AF	Ladurner [31]	Germany	07.19	Molecules	Prospective cohort	PTx/Tx	117	Endoscopy NIR camera (Karl Storz)	In vivo
AF	Thomas [33]	USA	09.19	Journal of the American College of Surgeons	Prospective cohort	PTx/Tx	20	NIR camera (PDE-Neo II) vs. PTeye	In and ex vivo
ICG + AF	Lerchenberger [32]	Germany	09.19	International Journal of Endocrinology	Prospective cohort	PTx/Tx	50	Endoscopy NIR camera (Karl Storz)	In vivo
AF	DiMarco [47]	UK	09.19	Annals of the Royal College of Surgeons of England	Prospective cohort	Tx	269	NIR camera (Fluobeam)	In vivo
AF	Liu [48]	China	01.20	BMC Surgery	Prospective cohort	Tx	20	i-Raman Pro NIR camera	In vivo
AF	Kose [34]	USA	01.20	Surgery (United States)	Prospective cohort	PTx/Tx	310	NIR camera (Fluobeam)	In and ex vivo
AF	Benmiloud [53]	France	02.20	JAMA Surgery	RCT	Tx	241	NIR camera (Fluobeam)	In vivo
AF	Idogawa [35]	Japan	05.20	European Archives of Oto-Rhino- Laryngology	Prospective cohort	PTx/Tx	45	NIR camera (PDE-Neo)	In vivo
AF	Serra [49]	Portugal	08.20	Gland Surgery	Prospective cohort	Tx	40	NIR camera (different fabricants)	Ex vivo
AF	Takahashi [12]	Japan	10.20	Laryngoscope	Prospective cohort	Tx	36	NIR camera (PDE-Neo)	In and ex vivo
AF	Kim [41]	USA	10.20	Journal of Surgical Oncology	Retrospective cohort	Tx	300	NIR camera (Fluobeam)	In and ex vivo
AF	Papavramidis [42]	Greece	01.21	Endocrine	RCT (single-blinded)	Tx	180	NIR camera (Fluobeam)	In and ex vivo
AF	Akbulut [36]	USA	03.21	Journal of Surgical Oncology	Prospective cohort	PTx/Tx	300	NIR camera (Fluobeam or Fluobeam LX)	-
ICG + AF	Barbieri [43]	Italy	07.21	Langenbeck’s Archives of Surgery	Prospective cohort	Tx	20	NIR camera (Fluobeam)	In vivo
AF	Bellier [44]	France	07.21	World Journal of Surgery	Prospective cohort	Tx	70	NIR camera (Fluobeam)	Ex vivo
AF	van Slycke [45]	Belgium	08.21	Surgical Innovation	Prospective cohort	Tx	40	NIR camera (Fluobeam)	In and ex vivo
AF	Kim [46]	South Korea	09.21	Thyroid: official journal of the American Thyroid Association	Retrospective cohort	Tx	542	NIR camera, unknown system	In and ex vivo
AF	Thomas [37]	USA	11.21	American Journal of Surgery	Prospective cohort	PTx/Tx	167	PTEye	In vivo
AF	Berber [38]	USA	12.21	Journal of Surgical Oncology	Retrospective cohort	PTx/Tx	239	Unknown device	Ex vivo

### 3.2. ICG

#### 3.2.1. Study Population

In the 24 studies that analyzed ICG-angiography, the number of included patients varied from 7 (proof-of-concept studies) to 210 (retrospective cohort) with a mean of 43.5 patients that underwent either total or partial thyroidectomy (*n* = 24). However, in four studies, both patients that underwent thyroidectomy and patients that underwent parathyroidectomy were included. Procedures were performed laparoscopically or by collar incision and in one case, transoral [56]. 

#### 3.2.2. Study Designs

Different study designs have been included, consisting mostly out of prospective cohort studies (*n* = 18 (75%)). Additionally, retrospective cohorts (*n* = 4), and randomized controlled trials (*n* = 3) were included (Table 2).

Several of these studies graded parathyroid ICG-perfusion based on a visual analysis by experienced surgeons (*n* = 8 (33%)) (Figure 2). These studies investigated the role of ICG in the identification of the parathyroid glands and predicted postoperative hypoparathyroidism based on the absence of at least one well-perfused gland. Vascularity was determined by a visual 2-ICG-scoring system for each identified parathyroid gland: 0 (no vascularity), 1 (medium vascularity) and 2 (excellent vascularity) [57,58,59].

Consequently, digital quantification of the fluorescent signal based on signal (parathyroid) to background (trachea) ratio was performed by Lang et al. This study attempted to quantify the ICG fluorescence signal in a single frame. Furthermore, it validated parathyroid tissue by biopsy instead of visually [60]. Some studies (*n* = 2 (8%)) compared autofluorescence and ICG-perfusion for intraoperative parathyroid identification and found similar intraoperative detection rates [32,51]. 

As an alternative to the 2-ICG-score, several studies investigated a 4-ICG-score (adding the individual 2-ICG-scores of 0, 1 or 2 of the 4 parathyroid glands, thus ranging from 0–8) to determine parathyroid viability and predict postoperative hypocalcemia [61,62]. A 4-ICG-score < 3 has a negative predictive value, positive predictive value, sensitivity and specificity for postoperative hypocalcemia of 95%, 42%, 83% and 73%, respectively, compared to 95%, 82%, 82% and 95%, respectively in the case of a single gland ICG-score of 2 [61]. For this reason, the 2-ICG-score appears to have higher diagnostic accuracy (92% vs. 72%), suggesting that one viable gland prevents (transient or permanent) postoperative hypoparathyroidism, however, the various expertise of the thyroid surgeon with the operative skills and the technique itself was not taken into account [63]. Several studies (*n* = 11 (46%)) validated their ICG-perfusion scores, either based on visual perfusion assessment by an experienced surgeon or based on intraoperative (*n* = 3 (13%)) or postoperative (*n* = 3 (13%)) PTH measurements (Figure 2) [56,57,64]. One study indicated a similar diagnostic accuracy of intraoperative PTH measurements and ICG perfusion assessment for the prediction of postoperative hypocalcemia (0.84 and 0.92, respectively) [65]. 

#### 3.2.3. Imaging Technique

Image acquisition was performed with a variety of NIR cameras or endoscopic/laparoscopic devices from varying companies. More frequently encountered are the Laparoscopic Pinpoint, NIR-endoscope (Karl-Storz) and NIR camera (SPY or Fluobeam). Only one study described the frame rate and gain [66]. Some studies describe quantification based on the heatmap camera mode, brightness and contrast settings (Table 2) [59,60,67]. 

#### 3.2.4. Standardization of Protocol

Studies have different priorities for the standardization of their protocols. When mentioned in articles (*n* = 3), camera angle and distance to the surgical field are perpendicular and 15–20 cm, respectively [60,66,68]. Non-laparoscopic NIR cameras are frequently positioned on a tripod, facilitating stabilization, thus maintaining a fixed distance to the surgical field [60,66]. None of the included studies mentions standardized procedures for the speed of ICG injection. Two studies do say that ICG has to be injected ‘’slowly’’, with one study stating that this allows better discrimination between the thyroid gland and the parathyroid tissue [56,65]. In general, there is heterogeneity in the total administered ICG dose with variations in frequency (in some studies this is mentioned as “multiple times”) and dose (2.5 mg–15 mg), but a general maximum of 5 mg/kg/day (Figure 2) [60,61,65,69]. 

Not all studies included a protocol for background light during the procedure. Two studies (8%) mention that the background light is switched off during the ICG procedure [31,66]. Only three studies (13%) quantified their fluorescence signal, of which two were postoperatively and one was intraoperatively (Figure 2). The quantification of fluorescence signals was attempted in two of these studies based on peak fluorescence intensity signal to background ratios. These studies revealed a lower TBR (0.6 vs. 5.1) in patients that developed postoperative hypocalcemia on the day after surgery and a significant correlation between fluorescent intensity and postoperative PTH (r = 0.598, *p* < 0.001) [60,70]. One recent article described quantification of the fluorescence intensity based on ICG inflow and outflow curve characteristics and showed that a combination of low ingress and egress slopes indicates parathyroid organ dysfunction in 100% of the cases [66]. Finally, there is a difference in the way the studies captured the images; in our analysis, five cases (21%) recorded a video of the ICG-angiography, three (13%) stored single images, and five (21%) described “recorded images” being taken. Remarkably, there were also eleven studies (46%) not describing whether images were saved, although in these cases no postoperative quantification of the signal was applied either.

#### 3.2.5. Study Outcomes

In general, there is a large variation in study outcomes. The use of ICG for identification and preservation alone was described as not significantly altering postoperative calcium and PTH levels [71]. Nonetheless, several prospective single-center proof-of-concept studies showed that intraoperative ICG perfusion imaging of the parathyroid glands was well tolerated, safe, reproducible and on top of that showed a strong correlation between perfusion and parathyroid function. However, they state the limitation that decision making is based on definitions of visual-perfusion-validation and ICG-perfusion scores [57,58]. 

Two RCTs that included patients undergoing thyroid surgery revealed that patients with at least one visually well-perfused parathyroid gland did not develop hypoparathyroidism and did not require postoperative calcium and vitamin D suppletion. This suggests ICG-angiography is a reliable predictor of parathyroid vascularization, removing the need for postoperative PTH and calcium measurements [62,69]. Opposing the last statement, a retrospective cohort study that included a total of 210 patients and investigated parathyroid gland evaluation based on visual perfusion and ICG-angiography perfusion scores showed no correlation between one well ICG-perfused parathyroid gland and postoperative PTH levels [72]. Additionally, a multicenter prospective study determined that ICG perfusion scoring did not seem to correlate with postoperative PTH (r = 0.011; *p* = 0.993) or serum calcium (r = 0.127; *p* = 0.335) [73]. On top of that, low-flow ICG patterns, meaning a vascularization score of zero, appeared not to be associated with postoperative changes in PTH levels and therefore, may lead to unnecessary autotransplantation [74]. 

With respect to the identification of the glands, ICG and autofluorescence show similar results, but autofluorescence more frequently identifies the glands before identification by the naked eye [32,51]. Concomitant use of autofluorescence, for parathyroid identification, and ICG imaging, for perfusion assessment, has been performed, but an effect on the incidence of postoperative hypoparathyroidism has not yet been demonstrated [27]. 

**Table 2 life-12-00388-t002:** Included articles utilizing ICG-angiography. AF= autofluorescence, ICG = indocyanine green, PTx = parathyroidectomy, Tx = thyroidectomy, NIR = near infra red.

Application	First Author	Origin	Publicated (MM.YY)	Journal	Study Design	Surgery	Sample Size	Imaging Technique
ICG	Fortuny [57]	Switzerland	04.16	British Journal of Surgery	Prospective cohort	Tx	36	Laparoscopy NIR camera (Pinpoint)
ICG	Zaidi [58]	USA	06.16	Journal of Surgical Oncology	Prospective cohort	Tx	27	NIR camera (Pinpoint)
ICG	Lang [60]	China	01.17	Surgery (United States)	Prospective cohort	Tx	70	NIR camera (SPY)
ICG	Yu [65]	Korea	07.17	Surgical Endoscopy	Prospective cohort	Tx	66	NIR camera (da Vinci Si robot system)
ICG + AF	Kahra-mangil [51]	USA	12.17	Gland Surgery	Prospective cohort	Tx	44	Pinpoint for ICG, Fluobeam for AF
ICG + AF	Alesina [27]	Germany	03.18	Langenbeck’s Archives of Surgery	Prospective cohort	PTx/Tx	5	Laparoscopy NIR camera (Karl Storz)
ICG	Fortuny [62]	Switzerland	03.18	British Journal of Surgery	RCT	Tx	196	Laparoscopy NIR camera (Pinpoint)
ICG	Jin [59]	China	12.18	Advances in Therapy	Prospective cohort	Tx	26	Digi-MIH-001 imaging system
ICG	van den Bos [70]	Netherlands	02.-19	Head and Neck	Prospective cohort	Tx	26	Laparoscopy NIR camera (Karl Storz)
ICG	Jin [67]	China	03.19	Clinical Endocrinology	Prospective cohort	Tx	26	Digi-MIH-001 imaging system
ICG	Rudin [72]	USA	06.19	World Journal of Surgery	Retrospective cohort	Tx	210	Laparoscopy NIR camera (Pinpoint)
ICG + AF	Ladurner [31]	Germany	07.19	Molecules	Retrospective cohort	PTx/Tx	117	NIR-endoscope (Karl Storz)
ICG	Razavi [74]	USA	09.19	Head and Neck	Retrospective cohort	Tx	43	NIR-endoscope (Karl Storz)
ICG + AF	Lerchenberger [32]	Germany	09.19	International Journal of Endocrinology	Prospective cohort	PTx/Tx	50	Unknown NIR camera + laparoscopy NIR camera (Olympus)
ICG	Gálvez-Pastor [61]	Spain	11.19	American Journal of Surgery	Prospective cohort	Tx	39	Laparoscopy NIR camera (Pinpoint)
ICG	Llorente [64]	Spain	01.20	JAMA Surgery	Prospective cohort	Tx	50	Unknown
ICG	Yavuz [68]	Turkey	04.20	Archives of Endocrinology and Metabolism	Prospective cohort	Tx	43	NIR camera (SPY)
ICG	Papavramidis [73]	Greece	09.20	Endocrine Practice	Prospective multicenter	Tx	60	NIR camera (OPAL)
ICG	Turan [56]	Turkey	10.20	Photodiagnosis and Photodynamic Therapy	Retrospective cohort	Tx/PTx	7	Laparoscopy NIR camera (Olympus)
ICG	Jin [69]	China	12.20	Endocrine Practice	RCT	Tx	56	Digi-MIH-001 imaging system
ICG	Llorente [63]	Spain	04.21	Cirugia Espanola	Prospective cohort	Tx	50	NIR camera (SPY)
ICG	Parfentiev [71]	Georgia	05.21	Georgian Med News	RCT	Tx	58	Laparoscopy NIR camera (Karl Storz)
ICG + AF	Barbieri [43]	Italy	07.21	Langenbeck’s Archives of Surgery	Prospective cohort	Tx	20	NIR camera (Fluobeam)
ICG	Noltes [66]	Netherlands	12.21	Annals of Surgery	Prospective multicenter	Tx	10	NIR camera (SPY)

### 3.3. Laser Speckle Contrast Imaging

#### 3.3.1. Selected Articles and Study Population

Two articles matching the inclusion criteria were found where LSCI was used to assess parathyroid perfusion. Both articles were written by the same American study group. The first study by Mannoh et al. (2017) was a feasibility study using a sample size of seven participants undergoing either a parathyroidectomy or a thyroidectomy [75]. The second study was an in-depth study of the first feasibility study containing a larger study population (*n* = 72) of patients undergoing a thyroidectomy (total or complementary) (Table 3) [76].

#### 3.3.2. Study Design

Both reported studies were prospective cohort studies, with the first study from 2017 mainly focusing on demonstrating the technique (*n* = 7), while the second study (2021) correlated the postoperative outcomes with the intraoperative LCSI measurements (*n* = 72). In the first study, measurements were validated by histological examinations on excised parathyroid glands in hyperparathyroidism patients. Since the second study was only performed in thyroidectomy patients, histopathological validation was not possible and a visual scoring by the surgeon was used as a measure of validation, although this could be considered an expert opinion instead of a validation. In both studies, the signal was quantified intraoperatively; however, the surgeon was blinded to these results.

#### 3.3.3. Imaging Technique

In the study from 2017, the LCSI method is described in detail. A basic LSCI setup is simple as this solely requires a camera, a laser source in the red or NIR range and diverging optics to generate the speckle pattern. The images are converted to 2D-perfusion maps by elementary mathematical equations, thus allowing for real-time quantification. The dye-free and full-field nature of the technology results in ease of use for parathyroid perfusion assessment. Mannoh et al. mention testing the influence of the overhead lights as well as the settings of the camera on the results [75]. Overhead light did not have any influence on the results. In the second study, the methods are mentioned briefly, and reference is made to the first article [76]. Both papers use the same LSCI device, although a slight change was made to the laser to reduce specular reflections in the images.

#### 3.3.4. Outcomes

The first paper showed the ability of this technique to distinguish between well and poorly vascularized parathyroid glands with high sensitivity (92.6%) and specificity (90.6%) [75]. In the 2021 article, the intraoperative measurements were correlated with postoperative outcomes, using the postoperative PTH value. This showed that intraoperative LSCI quantification can predict postoperative hypoparathyroidism with a sensitivity and specificity of 87.5% and 84.4%, respectively [76].

## 4. Discussion

This review aims to outline the developments in the application of autofluorescence, ICG and LCSI perfusion imaging in parathyroid-saving thyroid surgery in the last decade in order to investigate the possible reasons for previously reported heterogenous outcomes. In general, we found that general consensus in the applied protocols for the application of the different imaging techniques is lacking and is non-uniform in settings, execution, and analysis of optical imaging. Differences between studies using similar imaging techniques are found in, for example, the distance from the camera to the wound bed, the angle of view, the type of camera used, and, where applicable, the administered dose of ICG and its infusion protocol. Moreover, the majority of articles do not even mention such parameters at all. Part of the reviewed studies also omits the subsequent steps in the research process, such as feasibility studies that linked clinical consequences to the obtained results, while no validation of these results was done or reported.

When comparing the development of optical imaging with classical imaging techniques, such as magnetic resonance imaging (MRI), positron emission tomography (PET), and computed tomography (CT), optical imaging is still in the early development and clinical translation stages. In the clinical use of the current classical imaging techniques, like CT, MRI, ultrasound, or positron emission tomography (PET)-imaging, extensive research has been done into, for example, the pharmacokinetics and pharmacodynamics of contrast medium administration [77], resulting in standardized operating protocols for injection speed and length of infusion lines. Over the past decade, autofluorescence, ICG angiography and LSCI for the identification or preservation of parathyroid glands in thyroid surgery have developed rapidly, but these techniques are still in their early stages compared to classical imaging. Autofluorescence seems to be the least complicated, but most developed technique regarding standardization and quantification for this application (Figure 2). ICG angiography is also evolving rapidly, but in this field, there are so many applications, camera systems and protocols used, that the results of the different studies cannot be compared with each other. LSCI is the newest promising technique for the preservation of parathyroid glands in thyroid surgery, although only one study group published on this technique. Besides the two included articles, Mannoh et al. published a third interesting article on LSCI for preserving parathyroid glands in thyroid surgery, combining autofluorescence and LSCI in one device and validating their results with ICG-angiography. They demonstrated that LSCI can detect the perfusion of parathyroid glands similar to ICG-angiography [78]. However, this study only included three patients, of whom only one underwent thyroidectomy, and therefore this study was not analyzed in this review.

Although previous reviews on optical imaging in thyroid surgery have been published [15,16,79,80,81,82], these only focused on either the results or the imaging systems that were used. This review is the first to focus on the utilized methodology of the three aforementioned technologies in the included studies. Especially when analyzing the outcomes of studies reporting on a technical topic on fluorescent imaging, methodology homogeneity is crucial. Spartalis et al. published a review assessing the use of ICG-angiography for the identification of the parathyroid glands in both thyroid and parathyroid surgery. They came to the same conclusion as we did, namely that standardization and consensus are needed to implement this technique clinically [82]. However, in addition to the indication of their study being different from ours, they mainly looked at the administration protocol of the ICG, while we included a much wider range of standardization and quantification parameters, such as the use of the camera. Our analysis is the first to show that the methodology of all three technologies in the included studies is extremely heterogeneous. In our opinion, a worldwide consensus on the best method needs to be reached before a reliable meta-analysis can be performed. 

While in the field of autofluorescence and LSCI attention is paid to the quantification of the signal, this is not yet the case with ICG-angiography in total thyroidectomy. However, in other areas of interest, reviews on ICG-angiography concluded that time-related perfusion parameters, such as inflow parameters, provide a better predictive value for postoperative outcomes than the absolute fluorescence intensity [83,84]. The absolute fluorescence intensity was found to be strongly influenced by factors like the plasma ICG concentration, the amount of ambient light, the camera-to-wound bed distance, and the angle of the camera relative to the wound bed [84]. Furthermore, most authors omitted acquiring and storing images or videos of the ICG-angiography, therefore making it impossible to retrospectively correlate time-related parameters to postoperative outcomes, let alone to compare results in-between observers or in-between institutions. Moreso, the use of many different camera systems can be challenging in the process towards consensus for standardization and quantification. However, Ruiz et al. described a possible way to compare ICG-angiography between different cameras, using a standardized phantom with different fluorescence intensities for comparing the cameras [85]. This could be a step in the right direction in overcoming the differences in imaging between different cameras. 

Recently, our study group published a feasibility study in which a model (WISQ) was developed for the quantification of the fluorescence signal in parathyroid ICG-angiography. Postoperative hypoparathyroidism appeared to be well predicted by the combination of a poor ICG inflow curve and a poor outflow curve [66]. Further research on this proposed quantitative methodology for validation needs to be performed in the future. Ultimately, the goal should be to have a real-time model for determining the time-dependent fluorescence parameters with ease of use intraoperatively for the attending surgeon. With this in mind, a prediction for postoperative hypoparathyroidism may be made during surgery in order to support the clinical decision to autotransplant one or more parathyroid glands.

Despite the systematic methodology, this study has some limitations that should be mentioned. First of all, the use of ASReview makes screening more efficient, but may also lead to articles being missed since not literally every article is read or screened by a researcher. Second, there is no proper evaluation available of ASReviews error rate. However, screening by hand is also no impeccable method due to the possibility of human error resulting in accidentally missed articles, especially when many articles have to be screened. Moreso, ASReview has demonstrated to find 95% of eligible studies after screening 8–33% of studies by a researcher [18]. Furthermore, this study only investigated how the three different optical imaging techniques are applied for the identification and preservation of parathyroid glands in thyroid surgery. It should be noted that optical imaging of healthy parathyroid glands by means of autofluorescence cannot be compared with the use of the same technique for diseased parathyroid glands. In parathyroid adenomas, mixed high and low autofluorescence signals are seen [86]. Finally, this study included all articles that utilized LSCI, although there were only two of these published by the same research group. The lack of studies from multiple groups on this topic makes it difficult to evaluate the use and possibilities of standardization of this technique. 

## 5. Conclusions

The field of optical imaging-guided surgery for the localization and preservation of the parathyroid glands provides promising results. Nonetheless, standardization of the methodology is urgently required to make the next step to reproducible results into clinical practice. For future research, it is advisable to state and standardize the type of camera, camera settings, distance to the wound bed, the overhead light, the angle of view and, when applicable, the ICG dose and ICG injection protocols. Additionally, quantification models may provide a more objective interpretation of perfusion imaging. Altogether, these efforts might result in more homogeneous results and in the end facilitate incorporation of these techniques in a standard of care. Worldwide consensus on imaging protocols is needed to establish the clinical utility of these techniques for parathyroid gland identification and preservation.

## Figures and Tables

**Figure 1 life-12-00388-f001:**
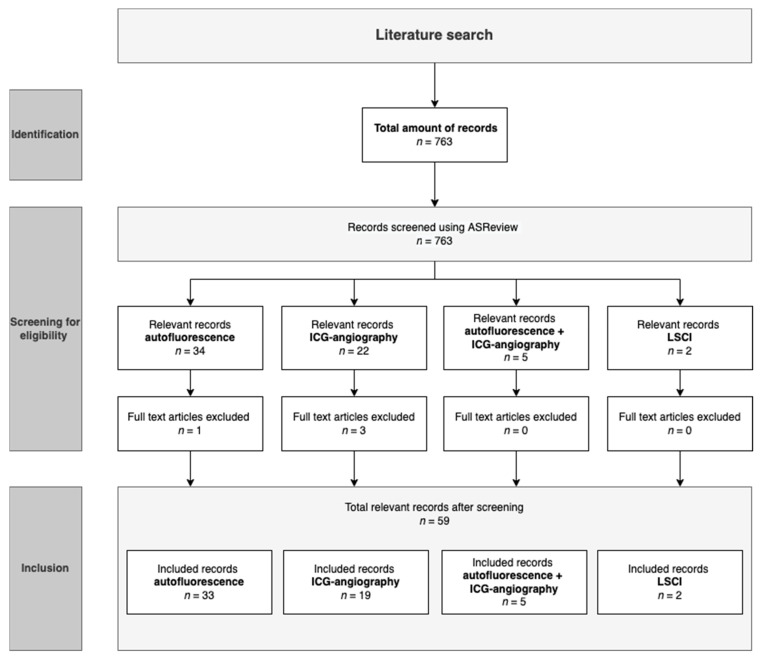
Process of article selection utilizing ASReview. LSCI = laser speckle contrast imaging, ICG = indocyanine green.

**Figure 2 life-12-00388-f002:**
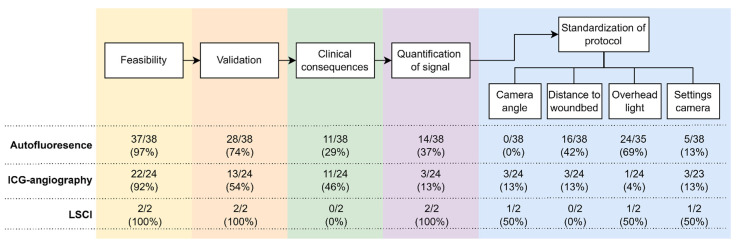
Flowchart showing the number of articles per stage ranged from weakest (**left**) to strongest (**right**). A study can have met multiple stages and has not necessarily completed all previous steps. ICG = indocyanine green, LSCI = laser speckle contrast imaging.

**Figure 3 life-12-00388-f003:**
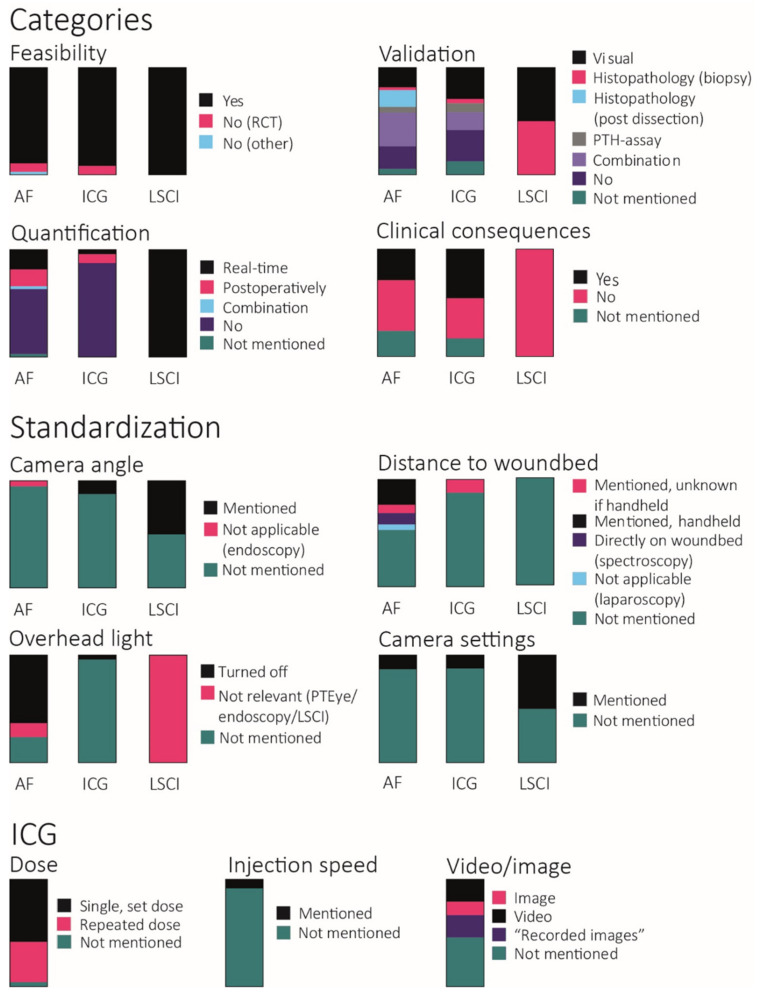
Variance in imaging protocols with bar graphs illustrating the number of articles that could be scaled in each category. Total amount: autofluorescence (AF) *n* = 38, ICG-angiography *n* = 24, laser speckle contrast imaging (LSCI) *n* = 2.

**Table 3 life-12-00388-t003:** Included articles utilizing laser speckle contrast imaging (LSCI). PTx = parathyroidectomy, Tx = thyroidectomy.

Application	First Author	Origin	Publicated (MM.YY)	Journal	Study Design	Surgery	Sample Size	Imaging Technique
LSCI	Mannoh [75]	USA	11.17	Scientific Reports	Prospective cohort	PTx/Tx	7	LSCI device
LSCI	Mannoh [76]	USA	10.21	Thyroid	Prospective cohort	Tx	72	LSCI device

## Data Availability

The full dataset is to be found in the Appendix A.

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
