# Peer review of "Heterogeneity in Utilization of Optical Imaging Guided Surgery for Identifying or Preserving the Parathyroid Glands—A Meta-Narrative Review"

_life, 2022, doi:10.3390/life12030388_

Round 1
Reviewer 1 Report
Fluorescence-guided surgery is a relatively new and powerful tool for all surgical specialties. This is a well written paper on the use of fluorescence-guided surgery on the identification and preservation of parathyroid glands during thyroid surgery. The work reviewed here represents the significant effort from many groups in the prevention of post thyroidectomy hypoparathyroidism via enhanced early identification, visualisation, and preservation of the parathyroid glands in neck endocrine surgery. The further development and standardisation of this promising technology will have a great impact on surgical outcomes.
However the main concern about this review is that there are already many (nearly 10) recently published similar and superposable reviews articles covering this exact topic (between others: 2021 - 34908194; 2021 - 34359693; 2017 - 28205245; 2020 - 31882459).
The authors need to clarify what is the relevance of writing another review article on this topic that doesn't seem to add any relevant and important information compared to previous works already published. We did not find significant added information or a new message that this paper would provide to the readership of Life.
A second concern is the lack of analyse concerning the identification and the appearance of diseased parathyroid glands (p.ex in hyperparathyroidism) that express a different autofluorescence appearance (less intense and heterogeneous) this point isn't even mentioned in the review but I think that should be absolutely analysed and reported.
Moreover, the introduction is too extensive and should be more succinct and focused. The same holds true for the discussion section.
References should be added in the tables.
minor comments
line 304 double spaced word "surgeon on"
Reviewer 2 Report
Very elaborate meta analysis, well structured and written review. In my eyes no revisions are needed, there is only one point where I've wondered a little. And I need to stress that it is of no real importance. In the part about auto fluorescence the authors mention geographic origins of the analysed publication, but the sum of mentioned publications doesn't add up to the total of 38. Those 3 missing publications from Argentina are missing.
Also in other sections (ICG etc.), no geographic distribution is mentioned. But I understand that these sections represent smaller bodies of publications.
Round 2
Reviewer 1 Report
Thank you for the review of the article. I respect the great effort for this kind of publications but the main concern remains and I still do not see this novelty/great added value of this literature review but this can be only a subjective point of view. At my knowledge the majority of the other reviews mention the heterogeneity of the technique as a limitation factor...however I agree that they did not systematically analyse it as you did.
In the introduction at least a mention to the different autofluorescence patterns in diseased PGs / false positives autofluorescence patterns(colloidal nodules brown fat exc ), I think should be done since is another source of heterogeneity...
